# Mitochondrial complex I deficiency occurs in skeletal muscle of a subgroup of individuals with Parkinson's disease
Simon Ulvenes Kverneng [1,2,3], Kjersti Eline Stige[1,2,3,4,5], Haakon Berven[1,2,3], Sepideh Mostafavi[1,2,3], Katarina Lundervold[1,2,3], Michele Brischigliaro [6,7], Brage Brakedal[1,2,3], Geir Olve Skeie[1,2,3], Irene Hana Flønes[1,2,3], Lilah Toker[1,2,3], Erika Fernandez-Vizarra[6,7], Ragnhild Eide Skogseth[8,9], Kristoffer Haugarvoll[1,2,3], Yamila N. Torres Cleuren[1,2,3], Christian Dölle [1,2,3], Gonzalo S. Nido[1,2,3] & Charalampos Tzoulis [1,2,3] ✉

## Abstract

**Background** Widespread neuronal mitochondrial complex I (CI) deficiency was recently reported to be a characteristic in a subgroup of individuals with idiopathic Parkinson's disease (PD). Here, we sought to determine whether a CI-deficient subgroup could be discerned using clinically accessible muscle biopsies. We further hypothesized that the inconsistency of previous findings of mitochondrial respiratory impairment in PD muscle may be due to interindividual variation, with respiratory deficiency only occurring in a subgroup of cases.

**Methods** Using a cross-sectional design, vastus lateralis needle biopsies were collected from 83 individuals with PD and 29 neurologically healthy controls and analyzed by immunohistochemistry for CI and complex IV (CIV), cytochrome c oxidase/succinate dehydrogenase (COX/SDH) histochemistry, and spectrophotometric activity assays of complexes I–IV. Mitochondrial DNA (mtDNA) copy number, deletions, and point variation were analyzed in single muscle fibers and bulk biopsy samples.

**Results** We show that PD muscle exhibits reduced CI activity at the group level, with 9% of cases falling below two standard deviations of the control group. In contrast, the activities of CII–CIV are not significantly different between the PD and control groups. No quantitative change of CI or CIV is detected, and the observed functional CI deficiency is not associated with mtDNA abnormalities.

**Conclusions** Our findings support the existence of a PD subpopulation characterized by CI pathology in skeletal muscle and suggest that stratification by extra-neural mitochondrial dysfunction may be informative for selecting individuals for clinical trials.

## Plain language summary

Parkinson's disease (PD) is a movement disorder in which nerve cells in the brain become damaged and die. One feature of the damaged cells in the brain is dysfunction of their microscopic energy factories called mitochondria. However, a recent study of brain tissue from people who died with PD found that only a subgroup of individuals showed widespread mitochondrial dysfunction. To determine whether a similar subgroup could be identified during life, we looked at samples from the muscles of people with PD and healthy people. We found dysfunctional mitochondria in muscles from some people with PD, but not all. Our work suggests that evaluating mitochondria in tissues outside of the brain may be useful for selecting people with PD who might benefit from treatments that target mitochondria.

Parkinson's disease (PD) is a clinicopathologically defined neurodegenerative disorder of unknown etiology[1]. It affects 1–2% of the population above the age of 65 years, and its prevalence is rapidly rising, making it one of the fastest-growing neurological diseases[2–4]. There are currently no disease-

modifying therapies for PD[5,6]. A significant barrier to mechanistic and therapeutic breakthroughs in PD is its heterogeneity. Individuals with PD demonstrate a wide range of clinical presentations, progression rates, and neuropathological signatures[7–9]. This phenotypical diversity has led to the

[1]Neuro-SysMed, Department of Neurology, Haukeland University Hospital, Bergen, Norway. [2]Department of Clinical Medicine, University of Bergen, Bergen, Norway. [3]K.G. Jebsen Center for Translational Research in Parkinson's Disease, University of Bergen, Bergen, Norway. [4]The Department of Neuromedicine and Movement Sciences (INB), Norwegian University of Science and Technology (NTNU), Trondheim, Norway. [5]Department of Neurology and Clinical Neurophysiology, St Olav's University Hospital, Trondheim, Norway. [6]Department of Biomedical Sciences, University of Padova, Padova, Italy. [7]Veneto Institute of Molecular Medicine, Padova, Italy. [8]Department of Geriatric Medicine, Haraldsplass Deaconess Hospital, Bergen, Norway. [9]Department of Clinical Sciences, Faculty of Medicine, University of Bergen, Bergen, Norway. ✉e-mail: charalampos.tzoulis@uib.no; charalampos.tzoulis@helse-bergen.no

https://doi.org/10.1038/s43856-025-00817-7                                                                                                **Article**

speculation that biological subtypes of PD, beyond the known monogenic forms, may exist, each with its own underlying mechanisms and therapeutic susceptibilities[6,10,11]. Identifying these subtypes will be pivotal for the development of targeted, disease-modifying interventions for PD[6].

Mitochondrial dysfunction has emerged as a central feature in the pathogenesis and pathophysiology of idiopathic PD[12]. This is supported by multiple findings, including evidence of compromised mitochondrial DNA (mtDNA) maintenance in the dopaminergic *substantia nigra pars compacta* (SNpc), and respiratory complex I (CI) deficiency in multiple brain regions, as well as extra-neural tissues of individuals with PD[12-15]. However, recent findings by our group suggest that such mitochondrial abnormalities are not universally present in PD. Rather, a pronounced and widespread neuronal CI deficiency occurs only in a subset of individuals, suggesting a distinct PD subtype, which we have termed CI-deficient PD (CI-PD). Based on our observations in post-mortem brain tissue, this group represents ~25% of idiopathic PD cases[16]. The CI-PD subtype may be particularly susceptible to therapeutic interventions targeting mitochondrial function, highlighting the need for stratification biomarkers for clinical trials.

In the current work, we hypothesized that it is possible to identify individuals with CI-PD through analysis of clinically accessible samples. This hypothesis was based on the observation that previous studies assessing mitochondrial function in peripheral tissue samples of individuals with PD have been highly inconsistent, with approximately half of the studies showing significant difference between PD and healthy controls at the group level[15]. While these conflicting findings could be a result of methodological differences and small sample sizes (<30 individuals with PD in most studies)[15], this observed variability raises the possibility that mitochondrial CI deficiency in extra-neural tissue is confined to the CI-PD subgroup. To address this question, we conducted a comprehensive analysis of the quantitative and functional integrity of the mitochondrial respiratory chain (MRC) in skeletal muscle biopsies of individuals with PD ($n = 83$) and neurologically healthy controls ($n = 29$). We chose skeletal muscle because of its high mitochondrial content and its status as a post-mitotic tissue that accumulates mitochondrial changes over time, similar to neurons[17]. Our study aimed to clarify the extent and nature of mitochondrial dysfunction in skeletal muscle of individuals with PD, potentially paving the way for new diagnostic and therapeutic approaches.

We show that a functional CI defect occurs in a subset of PD cases, accounting for 9–16% of our study cohort, depending on how the threshold of abnormal activity is defined. While not significant at the group level, some cases display reduced CII and CIV activities suggestive of a broader impairment of the MRC in PD muscle. The cause of the observed dysfunction is undetermined and not attributable to qualitative or quantitative alterations in mtDNA. Regardless of the cause, our findings suggest that PD may be stratified according to the presence of mitochondrial dysfunction in extra-neural tissue.

## Methods
### Cohort characteristics
Clinical data and skeletal muscle biopsies were collected from 83 individuals with PD participating in the NADPARK study[18] ($n = 25$) or the STRAT-PARK cohort[19] ($n = 58$), as well as 29 neurologically healthy controls participating in the STRAT-PARK cohort ($n = 23$) or the STRAT-COG cohort ($n = 6$). Study visits took place from January 2019 to February 2020 (NADPARK) and from December 2020 to January 2023 (STRAT-PARK and STRAT-COG) at the Department of Neurology, Haukeland University Hospital, Bergen, Norway (NADPARK and STRAT-PARK), the Department of Neurology and Clinical Neurophysiology, St. Olav's University Hospital, Trondheim, Norway (STRAT-PARK), and the Geriatric Medicine Outpatient Clinic, Haraldsplass Deaconess Hospital, Bergen, Norway (STRAT-COG). Only baseline samples were included from NADPARK participants who received active intervention. STRAT-PARK and STRAT-COG are longitudinal cohort studies of PD and dementia, respectively. Inclusion and exclusion criteria for the studies are provided in Supplementary Data 1. All individuals with PD had a clinical diagnosis of

established or probable PD, according to the Movement Disorders Society Clinical Diagnostic Criteria[1], as well as [$^{123}$I]FP-CIT single photon emission CT (DaTscan) with evidence of nigrostriatal degeneration. Because NAD-PARK participants were drug naïve at the time of biopsy, the clinical diagnostic criteria for these individuals were re-evaluated after initiation of dopaminergic treatment. Clinical assessment of individuals with PD included a medical history, a complete neurological examination, and the International Parkinson and Movement Disorder Society Unified Parkinson's Disease Rating Scale (MDS-UPDRS) parts I–IV[20]. Classification into tremor dominant (TD) and postural instability/gait difficulty (PIGD) phenotypes was based on MDS-UPDRS scores[21]. Additionally, Montreal Cognitive Assessment (MoCA) scores[22], occupational history, and self-reported data on pesticide exposure were available from STRAT-PARK participants. A conceptual overview of the study, including the experimental allocation of PD and control individuals, is provided in Fig. 1. Demographic and clinical characteristics (mean values) of the participants are provided in Table 1 and Supplementary Table 1. The study was approved by the Regional Committee for Medical and Health Research Ethics, Western Norway (NADPARK: 2018/597, STRAT-PARK: 74985, STRAT-COG: 216664). Written informed consent was obtained from all study participants.

### Skeletal muscle biopsy
A needle biopsy of the vastus lateralis muscle was performed using a Bard Magnum biopsy instrument (BD©, United States) with 12G x10 cm biopsy needle. Biopsies were dissected to remove non-muscle tissue (e.g., fat or fascia) before immediate freezing using isopentane pre-cooled in liquid nitrogen. Samples were stored at −80 °C until further analysis.

### Quadruple immunohistochemistry
The quadruple immunohistochemistry protocol was adapted from ref. 23. Using a cryotome (CM1950, Leica Biosystems), a transverse section of 12 μm thickness from each frozen biopsy was prepared onto glass slides and left to air dry for 60 min. Fixation was then achieved by immersion in 4% paraformaldehyde in PBS for 15 min at room temperature before rinsing with distilled water and permeabilizing and dehydrating by immersing in a series of methanol solutions (70% for 10 min; 95% for 10 min; 100% for 20 min, followed by 95% for 10 min; 70% for 10 min, and washing in TBS-T (0,1% Tween20 in TBS) for 5 min). Bovine serum albumin (4% in PBS) was then applied for 15 min to prevent non-specific binding. Subsequently, a cocktail of four primary antibodies diluted in TBS-T was applied, directed against VDAC1 (an outer mitochondrial membrane voltage-dependent channel, Abcam, #ab14734, dilution 1:100), NDUFB10 (a subunit of CI of the MRC, Abcam, #ab196019, dilution 1:200), MTCO1 (a subunit of CIV of the MRC, Invitrogen, #459600, dilution 1:200), and laminin (a basement membrane glycoprotein, Sigma–Aldrich, #L8271, dilution 1:1000). Staining was carried out in two batches, using the same working solutions of antibodies and reagents for all samples within the same batch. The samples in each batch were divided between three technicians who each included a negative control (no primary antibody). Primary antibodies were incubated for 1 h at room temperature. Following a washing step of TBS 2 × 5 min and TBST 1 × 5 min, a cocktail of four secondary fluorescent antibodies (Alexa Fluor™ 488 anti-mouse IgG2b, Invitrogen, #A-21141; Alexa Fluor™ 594 anti-rabbit IgG, Invitrogen, #A-11012; Alexa Fluor™ 647 anti-mouse IgG2a, Invitrogen, #A-21241; DyLight™ 405 anti-mouse IgG1, BioLegend, #409109) diluted 1:100 in TBS-T was applied and incubated in the dark for 1 h at room temperature. Sections were then washed in TSB-T 2 × 5 min and rinsed with TBS before mounting with ProLong™ Diamond Antifade Mountant (Invitrogen, #P36961). Following staining with secondary antibodies, sections were kept in the dark until image acquisition. For validation of the immunohistochemistry assay, a muscle biopsy was used from an individual with mitochondrial myopathy caused by the common ~5 kb single mitochondrial DNA (mtDNA) major arc deletion with known CI and CIV deficiency (Supplementary Data 2).

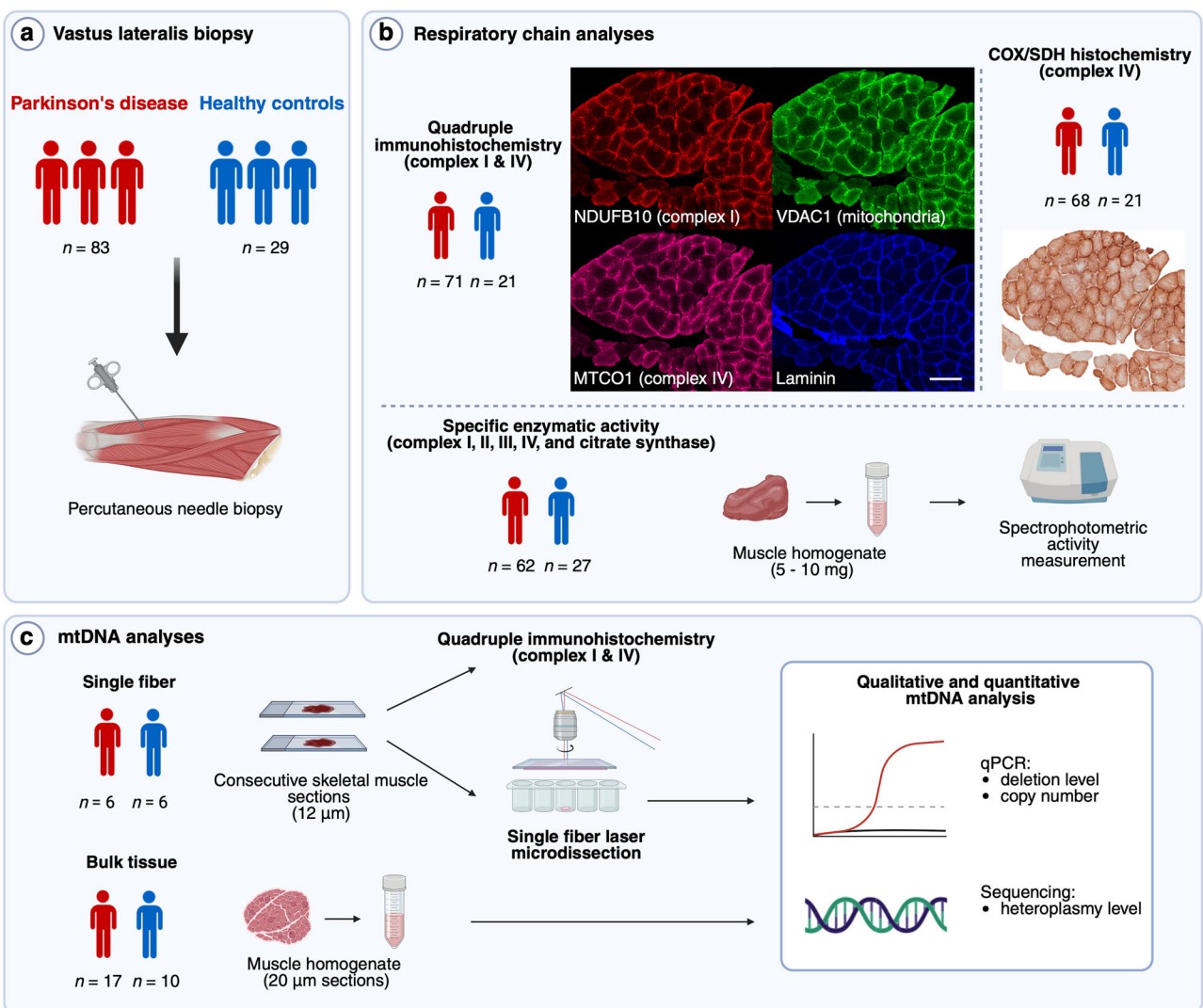

**Fig. 1 | Assessment of mitochondrial respiratory chain integrity in skeletal muscle biopsies from individuals with PD and neurologically healthy controls. a** Needle biopsies of the vastus lateralis muscle were collected from individuals with PD and neurologically healthy controls. **b** Respiratory chain analyzes: Quadruple immunohistochemistry was used for quantitative analysis of complexes I and IV (CI and CIV), the mitochondrial mass marker VDAC1, and for identifying single muscle fibers by laminin staining. Scale bar: 200 µm. Cytochrome c oxidase/succinate dehydrogenase (COX/SDH) histochemistry and spectrophotometric mitochondrial respiratory chain (MRC) complex activity assays were used for functional analysis of complexes I–IV (CI–IV) and citrate synthase (CS). **c** mtDNA analyses: The association between the quantity and function of CI and qualitative or quantitative changes in mtDNA was assessed. A subset of PD and control samples was selected for analysis. mtDNA copy number and deletions were assessed by quantitative PCR, and point variations were examined by PCR amplification and ultra-deep DNA sequencing, in both bulk muscle tissue and single muscle fibers. Figure created in BioRender. Kverneng, S. (2025) https://BioRender.com/m04e194.

## Image acquisition and fluorescence quantification

Fluorescent images were acquired at 40× magnification on a slide scanner (Olympus VS120 S6) and VS-ASW-S6 software, with a Hamamatsu ORCA-Flash 4.0. V3 B/W camera for fluorescence imaging, using a quad filter (DAPI, FITC, TRITC, and CY5, Supplementary Table 2) for fluor dyes at 405 nm (laminin), 488 nm (VDAC1), 594 nm (NDUFB10), and 647 nm (MTCO1). Exposure times were maintained for all channels between samples. Images were acquired at 16-bit. Image processing and quantification of fluorescence intensity were performed in ImageJ (ImageJ2, version 2.3.0/1.53f). Using the laminin signal, a mask was created to separate individual muscle fibers (Supplementary Fig. 1). In cases with poor laminin staining, the mask was created manually. Fluorescence intensity (mean gray value in the ImageJ software) was measured in 100 individual muscle fibers per muscle section, apart from seven sections where only 75–95 fibers were measured due to biopsy size. In addition, global fluorescence intensity was also measured in larger areas encompassing most of the muscle section (Supplementary Fig. 2).

Non-muscle tissue, such as blood vessels and connective tissue, was manually removed from the region of interest in a filtering step. The final region of interest was then generated by adjusting a threshold on the VDAC1 signal so that any defects in the tissue caused by thawing or sectioning were excluded. All fluorescence measurements were adjusted for background signal by subtracting the signal intensity acquired from a negative control belonging to the same batch and the same technician. Some sections were scanned twice due to technical adjustments of the slide scanner. This did not have a significant effect on fluorescence intensity levels when added to statistical models. The number of scans per section was therefore omitted as a variable from the final analysis of the immunohistochemistry dataset.

## Cytochrome c oxidase/succinate dehydrogenase histochemical staining

Cytochrome c oxidase/succinate dehydrogenase (COX/SDH) histochemical staining was used to identify COX-negative (CIV negative) muscle

## Table 1 | Demographic and clinical characteristics

| Variable | Group | |
| --- | --- | --- |
| | PD (*n* = 83) | Control (*n* = 29) |
| Sex (male/female) | 54/29 | 8/21 |
| Age (years) | 66.2 ± 7.4 | 65.4 ± 11.2 |
| MDS diagnosis (established/probable) | 76/7 | - |
| Disease duration (years) | 5.5 ± 4.8 | - |
| Motor phenotype: | TD: 40 PIGD: 33 IND: 10 | - |
| MDS-UPDRS III score | 28.0 ± 11.0 | - |
| Hoehn & Yahr stage (taken as part of the MDS-UPDRS) | 1.98 ± 0.6 | - |
| MoCA score | 25.7 ± 4.2[a] | 26.6 ± 2.3[a] |

*MDS diagnosis* Movement Disorders Society (MDS) Clinical Diagnostic Criteria for PD, *Disease duration (years)* duration of motor symptoms in years, *TD* tremor dominant, *PIGD* postural instability/gait difficulty, *IND* indeterminate, *MDS-UPDRS III* International Parkinson and Movement Disorder Society Unified Parkinson's Disease Rating Scale Part III, *MoCA* Montreal Cognitive Assessment.
Age, disease duration, MDS-UPDRS III score, Hoehn & Yahr stage, and MoCA score are presented as mean ± standard deviation.
[a]MoCA scores were available from 58/83 individuals with PD and from 23/29 controls.

fibers. In this assay, COX-positive fibers appear brown, while COX-negative fibers exhibit a blue stain due to intact SDH (CII) activity[24]. Transverse sections of 12 μm thickness were prepared as described above in the section on quadruple immunohistochemistry. Sections were left to air dry for 60 min before incubating with ~125 μl of COX-staining solution (prepared by combining 800 μl 5 mM diaminobenzidine tetrahydrochloride and 200 μl 500 mM cytochrome c mixed with a few grains of catalase) for 45 min at 37 °C. After washing in PBS for 3 × 5 min, sections were incubated with ~125 μl of SDH staining solution (prepared by combining 800 μl 1.875 mM nitroblue tetrazolium (NBT), 100 μl 1.3 M sodium succinate, 100 μl 2 mM phenazine methosulphate and 10 μl 100 mM sodium azide) for 40 min at 37 °C. Finally, sections were washed in PBS for 3 × 5 min before dehydration in a graded ethanol series (75% for 1 min, 95% for 1 min, and 100% for 10 min) and mounting. Brightfield images were acquired with a slide scanner (Olympus VS120 S6), and COX-status for single muscle fibers was qualitatively assessed (positive, intermediate, or negative) by two readers (SUK and CT).

### Mitochondrial respiratory chain enzymatic activity measurements

5–10 mg of a muscle biopsy was cut into small pieces using a surgical scalpel. The tissue was then homogenized in a glass-glass Dounce-type potter using 20 volumes of Medium A (0.32 M sucrose, 10 mM Tris-HCl pH 7.4, 1 mM EDTA) and 15 manual strokes. The homogenate was transferred to an Eppendorf tube and centrifuged at $800 \times g$ for 5 min at 4 °C. The pellet was discarded, and the supernatant was collected, transferred to a clean tube, and immediately frozen at −80 °C. These post-nuclear supernatants were kept at −80 °C overnight. Before performing the measurements, the samples were thawed at 37 °C and snap-frozen again in liquid nitrogen. This freeze-thawing cycle was carried out twice. The measurements of each of the respiratory chain complex activities and the citrate synthase (CS) activity were performed using 10–30 μl of mitochondria-enriched supernatant, in a total reaction volume of 200 μl, using a plate-reader spectrophotometer as described[25]. Briefly, rotenone-sensitive CI activity was measured by the addition of 0.2 mM nicotinamide adenine dinucleotide (NADH) and 50 μM coenzyme Q1 (CoQ1) and following the oxidation of NADH as a decrease in absorbance at $\lambda = 340$ nm, in both the absence and presence of 5 μM rotenone. Complex II (CII) activity was measured by following the reduction of 2,6-dichlorophenolindophenol (DCPIP; final concentration of 0.1 mM) as a change in absorbance at $\lambda = 600$ nm, in the presence of 1.5 mM KCN, 16 μM succinate, and 50 μM CoQ1. Complex III (CIII) activity was measured by

following the reduction of 50 μM cytochrome c as an increase in absorbance at $\lambda = 550$ nm, in the presence of 50 μM reduced decylubiquinone (DBH$_2$), prepared as described[25]. A baseline before the addition of the muscle sample was recorded to test for unspecific cytochrome c reduction. Complex CIV (CIV) activity was measured by following the oxidation of 90–95% reduced 100 μM cytochrome c as a decrease in absorbance at $\lambda = 550$ nm. The activities of CII, CIII, and CIV were assessed without the presence of specific inhibitors (malonate, antimycin A, and KCN, respectively) because only NADH oxidation by enzymatic activities other than CI generates a sufficiently high background signal to warrant the use of an inhibitor to determine specific CI activity. In contrast, the CII, CIII, and CIV activity signal is completely abolished upon addition of the respective specific inhibitor, indicating that any unspecific substrate conversion is negligible[25–27]. Eliminating these inhibitors allowed us to optimize the protocol by shortening assay time. The activity of CS was measured by following the change in absorbance at $\lambda = 412$ nm caused by the formation of thionitrobenzoate from 0.4 mM acetyl-CoA in a reaction initiated by 10 mM oxalacetate. Protein concentrations of the muscle homogenate samples were measured using the Bio-Rad Bradford protein assay. One PD sample and one control sample exhibited unusually high values of specific CS activity and were removed from the dataset as outliers (Supplementary Data 3).

### Laser microdissection and sample lysis

Six PD samples and six control samples were selected for laser microdissection of single muscle fibers to cover the spectrum of median single fiber VDAC1-adjusted CI level (Supplementary Fig. 3, Supplementary Data 4). Two serial sections of 12 μm were prepared from each biopsy. One was subjected to quadruple immunohistochemistry staining as described above, while the other one was placed on a membrane slide 2.0 μm PEN (Leica Microsystems, #11505158), air-dried for 30 min, and stained with hematoxylin before dehydration in ascending (70%, 95%, and 100%) ethanol solutions. Microdissection was performed on a laser microdissection microscope (Leica LMD7). Individual muscle fibers were collected in reaction tubes and lysed in 20 μl of lysis buffer (50 mM TrisHCl pH 8.0, 0.5% Tween20, 190 μg/mL proteinase K, diluted in sterile purified water) overnight at 56 °C. Samples were then centrifuged at $2000 \times g$ for 10 min at room temperature before incubation at 95 °C for 10 min to inactivate proteinase K. Finally, samples were cooled on ice before one last centrifugation step at $2000 \times g$ for 2 min at room temperature. During microdissection, the location of each dissected muscle fiber was recorded to allow for identification of the corresponding fiber on the serial immunostained section. In this manner, CI and CIV fluorescence quantification and mtDNA analyzes were obtained from the same single muscle fibers. Laser microdissection and immunohistochemistry were performed in three batches. One sample was included across all immunohistochemistry batches to calculate a between-batch correction factor for each of the three fluorescence targets of interest (i.e., NDUFB10, MTCO1, and VDAC1).

### Mitochondrial DNA copy number and deletion analysis in single muscle fibers

Determination of mtDNA copy number and major arc deletion fraction in microdissected single muscle fibers was performed using a duplex TaqMan quantitative PCR (qPCR) assay. The deletion fraction was determined by comparing a commonly deleted target (*MTND4*) to a rarely deleted target (*MTND1*)[28]. Deletion fractions in muscle samples were normalized to two blood genomic DNA samples from healthy controls, which were included in each experiment to serve as references for non-deleted mtDNA. Absolute quantification was carried out by comparison to a standard series containing PCR-amplified and purified *MTND1* and *MTND4* templates in known equimolar ($10^6$, $10^5$, $10^4$, $10^3$, and $10^2$ copies/μl). *MTND1* quantity was used to assess mtDNA copy number. For each muscle fiber, mtDNA copy number was calculated per microdissected area (μm$^2$). The following primers, probes, and conditions were used: *MTND1* forward primer: 5′-CCCTAAAACCCGCCACATCT-3′, *MTND1* reverse primer: 5′-GAGC-GATGGTGAGAGCTAAGGT-3′, *MTND1* MGB probe: 5′-FAM-

CCATCACCCTCTACATCACCGCCC-3′. *MTND4* forward primer: 5′-CCATTCTCCTCCTATCCCTCAAC-3′, *MTND4* reverse primer: 5′-CACAATCTGATGTTTTGGTTAAACTATATTT-3′, *MTND4* MGB probe: 5′-VIC-CCGACATCATTACCGGGTTTTCCTCTTG-3′. For each qPCR reaction, 2 μl of lysate was added to 18 μl of a master mix consisting of primers, probes, and TaqMan Advanced master Mix containing AmpliTaq® Fast DNA Polymerase (ThermoFisher) as per the manufacturer's instructions. Amplification was performed on a StepOnePlus™ Real-Time PCR System (ThermoFisher), using a thermoprofile of one cycle at 95 ℃ for 20 s, and 40 cycles at 95 ℃ for 1 s and 60 ℃ for 20 s. Each sample was run once in triplicate. Data were analyzed from a total of 111 muscle fibers from six individuals with PD and 112 muscle fibers from six controls (14–22 muscle fibers per individual).

## Mitochondrial DNA copy number and deletion analysis in bulk muscle tissue

After regressing out the effect of measurement batch, a total of $n = 27$ samples were selected based on CS-normalized CI activity: $n = 8$ PD samples with activity similar to controls, $n = 9$ PD samples with low activity, and $n = 10$ controls samples (Supplementary Fig. 4). These three groups were matched for age (Supplementary Table 1). A transverse section of 20 μm thickness was prepared from each muscle biopsy and collected in a 1.5 mL microtube, and DNA was extracted using the QIAamp® DNA Mini Kit (Qiagen). The resulting extracts were diluted in water to a DNA concentration of 1 ng/μL prior to qPCR. Determination of relative mtDNA copy number and the fraction of major arc deletion was performed in a triplex qPCR targeting *MTND1* and *MTND4*, as well as the nuclear gene *APP*. The following primers, probes, and conditions were used: *MTND1* forward primer: 5′-CCCTAAAACCCGCCACATCT-3′, *MTND1* reverse primer: 5′-GAGCGATGGTGAGAGCTAAGGT-3′, *MTND1* MGB probe: 5′-FAM-CCATCACCCTCTACATCACCGCCC-3′. *MTND4* forward primer: 5′-CCATTCTCCTCCTATCCCTCAAC-3′, *MTND4* reverse primer: 5′-CACAATCTGATGTTTTGGTTAAACTATATTT-3′, *MTND4* MGB probe: 5′-VIC-CCGACATCATTACCGGGTTTTCCTCTTG-3′. *APP* forward primer: 5′-TGTGTGCTCTCCCAGGTCTA-3′, *APP* reverse primer: 5′-CAGTTCTGGATGGTCACTGG-3, *APP* MGB probe: 5′-NED-CCCTGAACTGCAGATCACCAATGTGGTAG-3′. For each qPCR reaction, 2 μl of DNA sample was added to 18 μl of a master mix consisting of primers, probes, and TaqMan Advanced master Mix containing AmpliTaq® Fast DNA Polymerase (ThermoFisher) as per the manufacturer's instructions. Amplification was performed on a StepOnePlus™ Real-Time PCR System (ThermoFisher), using thermal cycling consisting of one cycle at 95 ℃ for 20 s, and 40 cycles at 95 ℃ for 1 s and 60 ℃ for 20 s. Samples were run three times in triplicate. Deletion fraction was determined as described above, while relative mtDNA copy number was calculated from the ratio of *MTND1* to *APP*[29]. A sample from blood genomic DNA was included in each experiment to serve as a non-deleted reference.

## Mitochondrial DNA sequencing

Sequencing of mtDNA was performed on DNA obtained from both microdissected single muscle fibers and bulk muscle tissue. mtDNA was amplified in two overlapping amplicons of 9307 bp (amplicon 1) and 7814 bp (amplicon 2), corresponding to positions 16,330 to 9068 and 8753 to 16,566 of the revised Cambridge reference sequence (rCRS), respectively. The following primers and conditions were used: 9307 bp amplicon forward primer: 5′-ACATAGCACATTACAGTCAAATCCCTTCTCGTCCC-3′, reverse primer: 5′-ATTGCTAGGGTGGCGCTTCCAATTAGGTGC-3′, 7814 bp amplicon forward primer: 5′-TCATTTTTATTGCCACAACTAACCTCCTCGGACTC-3′, reverse primer: 5′-CGTGATGTCTTATTTAAGGGGAACGTGTGGGCTAT-3′. For each of the PCR reactions, 2 μl of cell lysate was added to 18 μl of a master mix consisting of primers, dNTPs, 5X PrimeSTAR ® GXL buffer, and PrimeSTAR ® GXL (TaKaRa) DNA polymerase, per the manufacturer's instructions. Thermal cycling consisted of one cycle at 92 ℃ for 2 min and 40 cycles at 92 ℃ for 10 s, 65 ℃ for 30 s, and 68 ℃ for

8 min, as well as one final cycle at 68 ℃ for 8 min. Amplification was quality-controlled by gel electrophoresis using 4 μl of the PCR product. Samples with smearing of one or both amplicons were discarded. To establish the heteroplasmic call error introduced by PCR-amplification and DNA sequencing, the same fragments were amplified using a standard reference material mtDNA (Standard Reference Material® 2392-I, NIST, U.S. Department of Commerce) as template. The two amplicons from each sample were pooled prior to shipment to Novogene (UK) Co. Ltd for sequencing using the Illumina platform. The target depth for sequencing was set to 100,000× (~2 Gb per sample). The final mtDNA sequencing dataset consisted of (i) $n = 157$ single dissected muscle fibers from a total of 12 individuals (77 muscle fibers from 6 individuals with PD and 80 muscle fibers from 6 controls); (ii) one standard reference mtDNA control; (iii) $n = 27$ bulk muscle tissue samples from a total of 27 individuals (17 individuals with PD and 10 controls). For all samples, raw FASTQ sequencing files were trimmed to exclude low-quality bases and reads using Trimmomatic v0.39[30] with options: ILLUMINACLIP:TruSeq3-PE-2.fa:2:30:10 LEADING:3 TRAILING:3 SLIDINGWINDOW:4:15 MINLEN:36. Raw FASTQ files were assessed using FastQC[31] prior and following trimming. Reads were aligned to the hg38 human genome reference using BWA v0.7.17[32]. Reads mapping to the mitochondrial chromosome were then extracted and duplicates filtered out using GATK MarkDuplicates[33]. To allow for heteroplasmic genotyping, calling of variants was carried out using Mutserve v2.0.0-rc13[34]. To assess potential contamination, we calculated haplotype proportions in each sample using Haplocheck[35], which flagged 2/157 single-fiber samples with >1% contamination (30% and 15%). These fibers were discarded for downstream analyzes. Levels of contamination in bulk tissue samples were below 0.6%. Reads with mapping qualities below 20 were removed from the analyzes, and calling of heteroplasmy levels was restricted to the rCRS coordinates 600–8600 (amplicon 1) and 9100–16,200 (amplicon 2) to ensure adequate depth of coverage and potential sequences originating from the primers. Heteroplasmy levels were only considered if above 1% to ensure at least 10 reads covering the minor allele and restricted to single-nucleotide variants (i.e., deletions and insertions were removed). The heteroplasmy level was defined as the heteroplasmy of the minor allele for a given position. The two amplicon regions were analyzed separately since they exhibited different mean depth of coverage.

## Statistics and reproducibility

The sample size was not predetermined. Subjects were not randomized into experimental groups. Due to biopsy material limitations, histological assays and enzymatic activity assays were not performed in all study subjects. Sample sizes for each analysis are summarized in Supplementary Table 1. Histological staining was performed once for each biological sample. Enzymatic activity measurements were performed in technical triplicate for each sample. In the mtDNA qPCR assay, single muscle fiber samples were analyzed once in technical triplicate, while bulk muscle samples were analyzed three times in technical triplicate. mtDNA sequencing was performed once for each single muscle fiber sample and once for each bulk muscle tissue sample. One control individual was removed from the study due to suspicion of mitochondrial disease. One PD and one control sample were excluded from the enzymatic activity dataset due to unusually high CS activities (data provided in Supplementary Data File 3). Measurements from 15 samples were excluded from the analysis of complex III enzymatic activity due to technical issues with the reduction of decylubiquinone used in the assay. Investigators were not blinded during experiments or data analysis.

**COX/SDH histochemistry data.** Due to non-normality of the data, as judged by the Shapiro-Wilk test, the Wilcoxon rank-sum test with continuity correction was used to compare the individual proportions of COX negative or intermediate muscle fibers in the PD and control groups. Pearson's Chi-squared test was used to examine the association between disease state and the group-level proportion of individuals with any COX-negative or intermediate muscle fibers. Yate's continuity correction was applied as one expected cell frequency was below 10.

**Immunohistochemistry fluorescence intensity data.** All fluorescence intensity measurements were transformed using base 10 logarithms to improve model fits. To account for mitochondrial content variations, CI level was expressed as the ratio of NDUFB10-fluorescence to VDAC1-fluorescence, and CIV level as the ratio of MTCO1-fluorescence to VDAC1-fluorescence. Throughout the text, CI and CIV levels always refer to their VDAC1-adjusted levels, unless otherwise stated. Kendall's rank correlation coefficient was used to test the correlation between fluorescence intensity data and COX/SDH status. The correlation between fluorescence measurement in multiple single fibers and in a large section area was assessed by Spearman's Rank Correlation Coefficient, given as $\rho(n-2)$, due to non-normality as judged by the Kolmogorov–Smirnov test. Linear mixed effects regression models were used to compare the PD and control groups. Single muscle fiber VDAC1-fluorescence level, CI level, and CIV level were used as the dependent variables, while disease state, age, sex, smoking, and staining batch were included as fixed effects. Additionally, the study subject was added as a random effect to address inter-individual variability. Variance inflation factors (VIF) did not indicate collinearity. The analyzes were repeated in a dataset consisting of one large section area measurement per individual, using linear regression models with the same independent variables.

**Enzyme activity data.** To account for differences in mitochondrial content, the complex activities were normalized to the activity of CS. The resulting normalized activity measures (CI–CIV/CS), as well as CS activity alone, were log transformed using base 10 logarithms to improve model fits and used as dependent variables in linear regression models. The models included disease state, age, sex, and measurement batch as independent variables. Analyzes were repeated, adding smokers and including smoking as an additional variable. VIFs did not indicate collinearity. The comparisons of the PD and control groups for the four complexes were corrected for multiplicity using the Benjamini–Hochberg procedure. The analysis of CI/CS was also performed separately in the PD group, with MDS-UPDRS III score, MoCA score, disease duration, age, sex, and measurement batch as independent variables. The correlation between CI/CS and disease duration was tested using Spearman's Rank Correlation Coefficient, given as $\rho(n-2)$, due to non-normality as judged by the Shapiro–Wilk test. The correlations between CI/CS and CII/CS, CIII/CS, and CIV/CS within the PD group were assessed using Pearson's product-moment correlation, given as $r(n-2)$, since the Shapiro–Wilk test indicated normality. Log-transformed and batch-adjusted data were used for all correlation tests, and active smokers were omitted.

Comparisons between the PD subgroups with low and control-like CI/CS activity were conducted by Student's *t*-test for MoCA scores and age of onset, since the Shapiro–Wilk test indicated normality and the F-test indicated equality of variance. Wilcoxon rank-sum test with continuity correction was used to compare MDS-UPDRS III scores due to non-normality. Fisher's exact test was used to test the association between CI/CS subgroup and motor phenotype (tremor dominant, postural instability/gait difficulty) and sex. These analyzes were corrected for multiplicity using the Benjamini–Hochberg procedure.

**Adjustment of immunohistochemistry data and enzymatic activity data for batch effects.** For visualization and selected analyzes, immunohistochemistry fluorescence intensity data and enzymatic activity data were adjusted for the effects of batch. For the fluorescence intensity data (e.g., CI level), linear regression models were fitted using only staining batch as the independent variable. Similarly, for the enzymatic activity data (e.g., CI/CS), models were fitted using only measurement batch as the independent variable. In all instances, the dependent variable was log transformed using base 10 logarithms to improve model fits. The residuals from these regressions (i.e., variation in the data not explained by batch) were extracted, and the intercept of the regression was re-added to

maintain the original scale. The resulting batch adjusted data were used when plotting the distribution of fluorescence intensity data and enzymatic activity data; for calculating Cohen's *d* (determined by dividing the mean difference in CI/CS activity between the PD and control group by the pooled standard deviation); for the classification of the PD subgroups with low and control-like CI/CS activity; to assess the association between CI/CS activity and CI level; for correlation between CI/CS and CII/CS, CIII/CS, and CIV/CS in the PD group; for correlation between CI/CS and disease duration; and for the selection of samples for mtDNA analysis in single fibers and bulk tissue.

**Association between complex I enzyme activity data and immunohistochemistry data.** The association between CI activity and quantity was assessed a linear regression model with CI/CS as dependent variable, and disease state, age, sex, and CI level (NDUFB10/VDAC1) as independent variables, using fluorescence measurements from large section areas. Batch-adjusted data was used, and both CI/CS activity data and CI level data were transformed using base 10 logarithms. To assess whether including CI level data significantly improved model fit, an analysis of variance (ANOVA) was conducted using the *anova* function in R to compare the model with and without CI level as a covariate.

**mtDNA data.** Single fiber mtDNA copy number and major arc deletion fraction were assessed in linear mixed effects regression models that included disease state, age, sex, and qPCR plate as fixed effects. The study subject was added as a random effect variable to account for inter-individual variability. VIFs did not indicate collinearity. Due to non-normality of the data as judged by the Kolmogorov–Smirnov test and the Shapiro–Wilk test, the correlations between mtDNA parameters and immunohistochemical fluorescence intensity measurements were assessed by Spearman's Rank Correlation Coefficient, given as $\rho(n-2)$. In bulk muscle tissue, due to non-normality of the data as judged by the Shapiro–Wilk test, the Wilcoxon rank sum test with continuity correction was used to compare mtDNA copy number and deletion fraction between the PD group with CI activity similar to controls and the PD group with low CI activity, as well as between each PD group and the control group. For single fibers, the heteroplasmic load was modeled as the sum of the heteroplasmic levels (above 1%) across the entire amplicon region as a function of disease state, age, sex, and single fiber CI level, accounting for individual as a random effect using linear mixed effects models. For bulk muscle tissue samples, the same measure of heteroplasmic load was employed (i.e., the sum of the heteroplasmic levels). To assess the association between heteroplasmic load and disease status (i.e., PD-normal CI activity, PD-low CI activity, and control), the Kruskal–Wallis test was used, due to non-normality of the data as assessed by the Kolmogorov–Smirnov test. The association between heteroplasmic load and CI activity was then assessed, with a linear regression model with log(CI/CS) as the independent variable and accounting for activity measurement batch.

All analyzes were performed using R version 4.3.0 (R Core Team, 2023) in RStudio 2023.03.1 Build 446 (2009–2023 Posit Software, PBC). Linear mixed effects regression models were done using the *lme4* package V1.1.35.1[36]. Regression model summary tables were obtained using the *tab_model* function of the *sjPlot* package V2.8.16[37]. The *ggplot2* package V3.4.4 was used for plots[38]. Adjusting data for the effect of batch, as described above, was achieved using the *adjust* function of the *datawizard* package V0.9.1[39]. Calculation of power and sample size was performed using *pwr* V1.3.0 and *rstatix* V0.7.2[40,41]. A *P* value < 0.05 was considered statistically significant. All tests were two-sided.

**Reporting summary**
Further information on research design is available in the Nature Portfolio Reporting Summary linked to this article.

**Table 2 | Linear mixed effects models of CI and CIV levels in single muscle fibers in individuals with PD and controls**

| | Dependent variable | | | | | |
|---|---|---|---|---|---|---|
| | log(NDUFB10/VDAC1) | | | log(MTCO1/VDAC1) | | |
| Predictors | B | 95% CI | P value | B | 95% CI | P value |
| Disease state (PD) | −0.016 | −0.047 to 0.014 | 0.290 | −0.010 | −0.050 to 0.029 | 0.609 |
| Age | −0.002 | −0.003 to −1.2e−04 | **0.034** | −0.002 | −0.004 to 2.5e−04 | **0.026** |
| Sex (Male) | 0.019 | −0.007 to 0.045 | 0.153 | 0.052 | 0.018–0.086 | **0.003** |
| Smoking | −0.005 | −0.057 to 0.047 | 0.849 | −0.020 | −0.089 to 0.048 | 0.564 |
| Batch (Batch 2)[a] | −0.102 | −0.126 to −0.078 | **6.9e−17** | 0.064 | 0.032 to 0.095 | **7.0e−05** |
| Random Effects | | | | | | |
| $\sigma^2$ | 0.004 | | | 0.007 | | |
| $\tau_{00\ Subject}$ | 0.003 | | | 0.006 | | |
| ICC | 0.479 | | | 0.439 | | |
| $n_{Subject}$ | 92 | | | 92 | | |
| Observations | 9073 | | | 9073 | | |
| Marginal $R^2$/Conditional $R^2$ | 0.305/0.638 | | | 0.107/0.499 | | |

log(NDUFB10/VDAC1) complex I level adjusted for mitochondrial content, log(MTCO1/VDAC1) complex IV level adjusted for mitochondrial content, B regression coefficient (unstandardized), 95% CI 95% confidence interval of the regression coefficient, $\sigma^2$ residual variance, $\tau_{00\ Subject}$ random intercept variance, ICC intraclass correlation coefficient, representing the proportion of total variance in the dependent variable attributable to the grouping structure (i.e, subjects), $n_{Subject}$ number of study subjects.
Significant P values are in bold. Nominal P values are given.
[a]Immunohistochemistry staining was performed in two batches.

## Results

### Validation of quadruple immunohistochemistry for quantitative mitochondrial respiratory chain assessment

Quantitative MRC assessment in muscle was performed by quadruple immunohistochemistry for CI, CIV, VDAC1, and laminin. CI and CIV levels were normalized to VDAC1 to account for total mitochondrial mass. To assess the validity of our assay, we compared it to COX/SDH histochemical staining in serial sections from a muscle biopsy of an individual with mitochondrial myopathy caused by the common ~5 kb single mtDNA major arc deletion, which exhibited multiple COX-negative fibers (Supplementary Fig. 5, Supplementary Data 2). There was a strong correlation between COX positivity and CIV levels in single muscle fibers as assessed by immunohistochemistry (Kendall's $\tau = 0.71$, $n = 60$, $P = 7.81 \times 10^{-12}$, Supplementary Fig. 5h). While there is no reliable histochemical assay to evaluate CI integrity, fibers with CIV deficiency due to mtDNA deletion are often also deficient for CI[42]. In line with this, there was a significant correlation between COX positivity and CI level (Kendall's $\tau = 0.71$, $n = 60$, $P = 7.04 \times 10^{-12}$, $n = 60$, Supplementary Fig. 5i).

### Cytochrome c oxidase/succinate dehydrogenase histochemistry shows no complex IV deficiency in PD muscle

Skeletal muscle sections from 68 individuals with PD (46 males, 22 females, mean age 65.9 ± 7.8 years) and 21 neurologically healthy controls (5 males, 16 females, mean age 61.1 ± 9.9 years) were assessed by COX/SDH histochemical staining (Supplementary Fig. 6). The demographic and clinical information of the study cohort is shown in Table 1, while experimental allocation and demographic and clinical information per analysis is shown in Supplementary Data 5 and Supplementary Table 1, respectively. There were generally few COX-negative fibers, ranging between 0 and 7 per section (Supplementary Data 6). One control individual had 23 COX-negative fibers, which raised suspicion of mitochondrial disease. This individual was excluded from the study. The proportions of COX-negative or intermediate muscle fibers per section were not significantly different in the PD and control groups (Supplementary Table 3). Similarly, there was no significant difference between the PD and control groups in the proportion of individuals with any COX-negative or intermediate muscle fibers (Supplementary Table 3).

### Immunohistochemistry shows no quantitative changes in complexes I or IV in PD muscle

We next implemented the quadruple immunohistochemistry assay to assess the level of CI and CIV in skeletal muscle sections from 71 individuals with PD (47 males, 24 females, mean age 66.2 ± 7.8 years) and 21 neurologically healthy controls (5 males, 16 females, mean age 61.1 ± 9.9 years; Supplementary Data 5). In each section, the fluorescence signal was measured in single muscle fibers ($n = 75$–100 per section) to capture inter-fiber variability, as well as in a single large area encompassing the majority of muscle fibers of the section (Supplementary Fig. 2). The two approaches showed a high level of correlation at the subject level for both CI (Spearman's $\rho(90) = 0.97$, $P = 2.2 \times 10^{-16}$) and CIV (Spearman's $\rho(90) = 0.95$, $P = 2.2 \times 10^{-16}$; Supplementary Fig. 7).

Single fiber data were then analyzed using a linear mixed effects model (LMM) with disease state, age, sex, smoking status, and staining batch as fixed effects, and study subject as a random effect, with the dependent variable log-transformed (Table 2 and Supplementary Data 4). At the group level, muscle fibers from cases and controls displayed similar CI and CIV levels (Fig. 2a–d and Supplementary Fig. 8, Table 2). Age was negatively associated with both CI levels (LMM, $B = -0.002$, $P = 0.034$, Table 2) and CIV levels ($B = −0.002$, $P = 0.026$, Table 2). While we did not observe an effect of sex on CI levels, male subjects exhibited significantly higher CIV levels (LMM, $B = 0.052$, $P = 0.003$, Table 2). Single fiber VDAC1-fluorescence showed similar distribution and levels in the PD and control groups (Fig. 2e, f), indicating similar mitochondrial content, and was not significantly associated with age, sex, or smoking status (Supplementary Table 4). Analysis of the measurements taken from large areas of the sections, encompassing the majority of muscle fibers from each sample, yielded similar results (Supplementary Table 5 and Supplementary Data 7).

**Complex I activity is reduced in PD muscle.** The specific enzymatic activities of CI–CIV and CS were measured in muscle biopsy samples. Active smokers were omitted because smoking is known to inhibit mitochondrial respiration[43], and in particular CI function[44–47]. After removal of two outliers (Methods; Supplementary Data 3), the dataset consisted of 57 individuals with PD (35 males and 22 females, mean age 67.1 ± 7.4 years) and 25 controls (8 males and 17 females, mean age

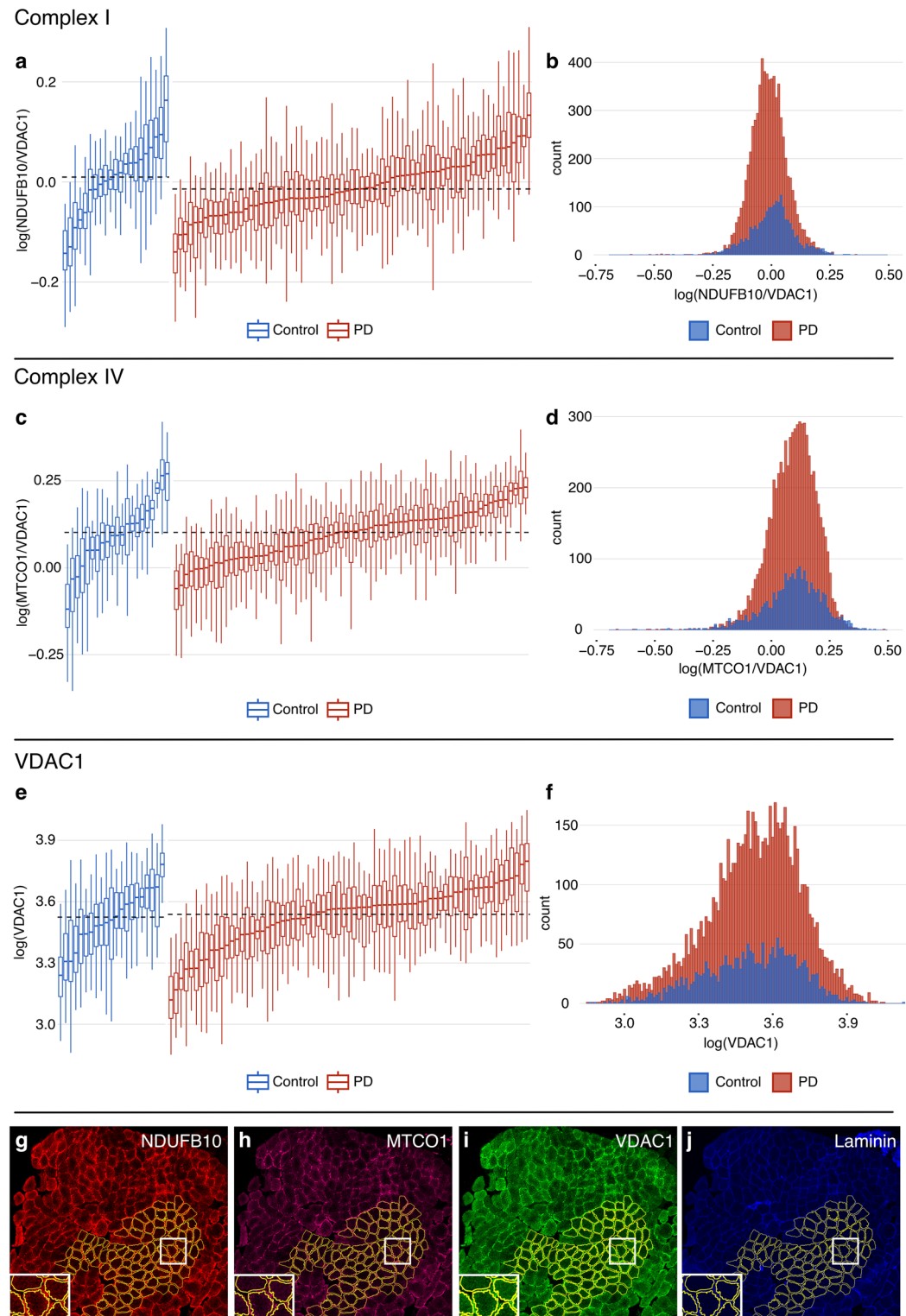

**Fig. 2 | Immunohistochemical assessment of complexes I and IV in PD single muscle fibers.** Complex I (NDUFB10); (**a**, **b**) and complex IV (MTCO1); (**c**, **d**) fluorescence intensity normalized to mitochondrial mass (VDAC1) in single muscle fibers ($n = 75–100$ per individual) from 71 individuals with PD (red) and 21 neurologically healthy controls (blue). VDAC1 fluorescence intensity measurements from the same fibers are shown (**e**, **f**). Values are log transformed. For the purpose of visualization, the data have been adjusted for the effect of staining batch by regressing out this variable (see "Methods" section). Boxplots (**a**, **c**, **e**) show individual-level distributions of single fiber measurement where each box represents one individual. Boxes: median and interquartile range (IQR); whiskers: 1.5 x IQR from the lower and upper quartiles. Individuals are sorted by median values from left to right. Dashed lines show the group-level medians of the PD and control groups. The histograms (**b**, **d**, **f**) represent group-level distributions of single fiber measurement in the PD and control groups. Supplementary Fig. 8 shows complex I and complex IV data without adjusting for staining batch. A representative example of quadruple immunohistochemistry in a skeletal muscle biopsy from an individual with PD shows fluorescence signal for NDUFB10 (**g**), MTCO1 (**h**), the mitochondrial mass marker VDAC1 (**i**) and laminin (**j**). Yellow borders represent regions of interest (ROIs) used for measuring fluorescence signal in 100 single muscle fibers. Zoomed-in regions are indicated by white squares. Scale bar: 200 μm.

**Table 3 | Linear regression models of enzymatic activity of CI, II, III, IV, and CS in individuals with PD and controls**

| Predictors | log(CI/CS) | | | log(CII/CS) | | | log(CIII/CS) | | | log(CIV/CS) | | | log(CS) | | |
|---|---|---|---|---|---|---|---|---|---|---|---|---|---|---|---|
| | B | 95% CI | P value (adjusted)[a] | B | 95% CI | P value (adjusted)[a] | B | 95% CI | P value (adjusted)[a] | B | 95% CI | P value (adjusted)[a] | B | 95% CI | P value |
| Disease state (PD) | −0.079 | −0.137 to −0.021 | **0.008 (0.032)** | −0.043 | −0.103 to 0.018 | 0.163 (0.313) | −0.038 | −0.102 to 0.025 | 0.235 (0.313) | −0.027 | −0.089 to 0.035 | 0.383 (0.383) | 0.034 | −0.090 to 0.159 | 0.582 |
| Age | −0.001 | −0.004 to 0.003 | 0.693 | 0.002 | −0.001 to 0.006 | 0.143 | 0.008 | 0.005 to 0.011 | **1.3e−05** | 0.003 | −1.8e−04 to 0.007 | 0.063 | −0.001 | −0.008 to 0.006 | 0.826 |
| Sex (Male) | 0.027 | −0.027 to 0.082 | 0.323 | 0.013 | −0.043 to 0.070 | 0.640 | 0.006 | −0.052 to 0.064 | 0.846 | 0.017 | −0.041 to 0.076 | 0.555 | −0.023 | −0.140 to 0.093 | 0.690 |
| Batch (1−6)[b] | - | - | - | - | - | - | - | - | - | - | - | - | - | - | - |
| Observations | 82 | | | 82 | | | 67[c] | | | 82 | | | 82 | | |
| R²/R² adjusted | 0.481/0.424 | | | 0.777/0.752 | | | 0.762/0.734 | | | 0.348/0.276 | | | 0.363/0.293 | | |

CI complex I activity, CII complex II activity, CIII complex III activity, CIV complex IV activity, CV complex IV activity, CS citrate synthase activity, x/CS activity x (CI, CII, CIII, or CIV) normalized to citrate synthase activity, B regression coefficient (unstandardized), 95% CI 95% confidence interval of the regression coefficient.
Significant P values are in bold. Nominal and adjusted P values are given.
[a] Parentheses show P values adjusted for multiple testing using the Benjamini-Hochberg procedure for four tests, i.e., CI/CS, CII/CS, CIII/CS, and CIV/CS between the PD and control groups.
[b] Detailed coefficients for batch variables are provided in Supplementary Table 6.
[c] Complex III activity measurements from 15 individuals were excluded from the analyzes due to technical issues with the reduction of decylubiquinone.

65.4 ± 12.0 years; Supplementary Data 5). Data were analyzed using a linear regression model with disease state, age, sex, and measurement batch as independent variables, and with the dependent variable log-transformed (Table 3 and Supplementary Data 3). CS activity, reflecting mitochondrial mass, showed no significant difference between individuals with PD and controls or association with any of the covariates. In contrast, CI activity normalized by CS activity (CI/CS) was significantly lower in the PD group compared to the controls (linear regression, B = −0.079, P = 0.008), and remained significant after controlling for multiple testing (P = 0.032), while the CS-normalized activities of CII–IV (CII–IV/CS) showed no significant differences (Fig. 3a–d, Supplementary Fig. 9, Table 3, Supplementary Table 6). The MRC complex activities were not influenced by age or sex, with the exception of CIII, which showed a positive association with age (linear regression, B = 0.008, P = 1.3 × 10⁻⁵, Table 3). The regression coefficient was back-transformed from log for interpretation. This revealed a 17% lower CI/CS in the PD group compared to controls, corresponding to a medium effect size (Cohen's d) of 0.65. Similar results were observed when smokers were included in the analysis (Supplementary Figs. 10–11 and Supplementary Table 7), although in this case the difference in CI/CS between the PD and control group did not remain significant after controlling for multiple testing.

Since previous results in PD muscle have been conflicting, we performed a power calculation based on our observed effect size (Cohen's d) of 0.65 to estimate the power of discovery as a function of sample size. This showed that using a Student's t-test, a total sample size of n = 78 or n = 103 would be required to detect a significant difference between cases and controls at a two-sided significance level of 5% and power of 80% or 90%, respectively. A plot of power as a function of sample size is shown in Supplementary Fig. 12.

CI/CS activity data were further analyzed within the PD group to investigate associations with disease severity. There was no correlation between disease duration and CI/CS activity (Spearman's ρ(55) = −0.05, P = 0.689; Fig. 3e). Furthermore, linear regression revealed no association between CI/CS activity and the MDS-UPDRS part III score, MoCA score, disease duration, sex or age (Supplementary Table 8).

**Reduction of complex I activity is not pervasive in PD.** We next investigated the distribution of CI/CS activity to determine the pervasiveness of CI deficiency in PD muscle. Plotting the activity after regressing out the batch effect revealed a substantial overlap between the PD and control groups (Fig. 3a). Out of 57 individuals with PD, 9 (15.8%) displayed CI/CS activity below the range of controls. Using a more stringent criterion, whereby abnormal CI/CS activity was defined as falling below 2 standard deviations of the control group, 5 out of 57 (8.8%) were classified as CI deficient. These individuals showed a mean decrease in CI/CS activity by 24.5% (range 17.8–29.1%) compared to the mean of controls.

Compared to the subgroup with CI/CS activity within the range of controls, the group with low CI/CS activity showed a significant female preponderance (Fisher's exact, P = 0.020; Fig. 3f). However, this did not remain significant after adjusting for multiple testing. There were no differences between the subgroups in terms of age of onset, cognitive function measured by MoCA, or motor function measured by MDS-UPDRS III. Moreover, there was a similar proportion of cases with tremor dominant and postural instability/gait difficulty phenotypes in both groups (Fig. 3g). None of the individuals with activity below the range of controls reported agricultural work, while a single individual reported previous pesticide exposure. These results are summarized in Supplementary Table 9.

In light of the CI/CS activity findings, the CI immunohistochemistry data were re-analyzed, including only subjects from whom both activity and immunohistochemistry data were available (45 individuals with PD and 17 controls). At the group-level, there was still no significant difference in CI levels between the PD and control group (Supplementary Table 10). To assess whether the decrease in CI/CS activity could be explained by

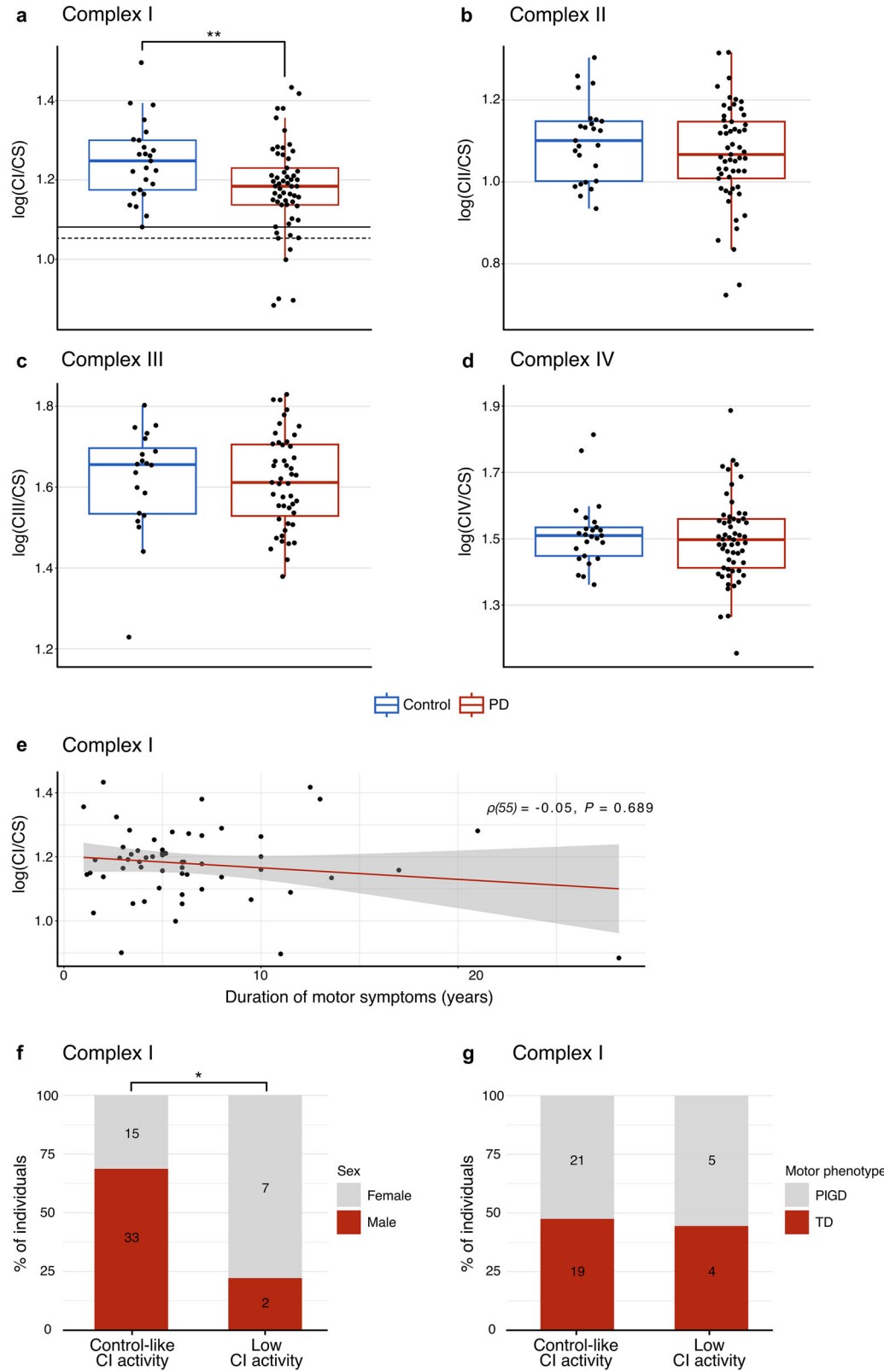

a reduction in CI quantity, we compared the goodness of fit of a model predicting CI/CS activity with and without including the CI immunohistochemistry data. ANOVA between the two models indicated that inclusion of the CI quantity data significantly improved the goodness of fit of the model ($P = 0.002$, Supplementary Table 11). Furthermore, there was a highly significant positive association between CI activity and quantity (linear regression, $B = 0.806$, $P = 0.002$, Supplementary Table 12).

**Single muscle fiber mitochondrial DNA profile shows no difference between PD and controls.** Next, we assessed whether PD muscle harbored qualitative or quantitative changes in mtDNA at the single fiber

**Fig. 3 | Spectrophotometric activity measurement of mitochondrial respiratory complexes in PD muscle.** Activities of complexes I–IV (CI–IV), normalized to citrate synthase (CS) activity, in muscle biopsies from 57 individuals with PD and 25 neurologically healthy controls, are shown. Values are log transformed. For the purpose of visualization, the data have been adjusted for the effect of measurement batch by regressing out this variable (see "Methods" section). Smokers have been removed from the dataset (Supplementary Figs. 9–11 show **a**–**d** without adjusting for batch effects and including active smokers). Red boxplots represent the PD group and blue boxplots represent the control group. Boxes: median and interquartile range (IQR); whiskers: 1.5 x IQR from the lower and upper quartiles. Each dot represents one individual. **a** CS-normalized CI activity. The solid line indicates the lower end of the control distribution. The dotted line indicates minus 2 standard deviations from the mean of controls. Statistical significance was determined using a linear regression model. **b** CS-normalized CII activity. **c** CS-normalized CIII activity. Measurements from 15 individuals were excluded from this analysis due to technical issues with the reduction of decylubiquinone (Supplementary Data 5). **d** CS-normalized CIV activity. **e** Correlation between duration of motor symptoms and CS-normalized CI activity in the PD group. The red line represents the linear regression line, and the gray shaded area shows the 95 % confidence band. Stacked bar plots show the proportion (in percentage) of males/females (**f**) and TD/PIGD motor phenotype (**g**) in the PD subgroups with CS-normalized CI activity within (control-like CI activity) and below (low CI activity) the range of the control group. Counts are displayed in the bars. Statistical significance was determined by Fisher's exact test (two-sided). *TD* tremor dominant *PIGD* postural instability/gait difficulty. *: $P = 0.020$; **: $P = 0.008$.

level. A total of 223 single muscle fibers were analyzed from six individuals with PD (5 males and 1 female, mean age $66.8 \pm 6.5$ years), spanning the CI level range, and six controls (1 male and 5 females, mean age $64.8 \pm 9.5$ years; Supplementary Fig. 3, Supplementary Data 5). In each fiber, CI and CIV levels were determined using quadruple immunohistochemistry, and mtDNA copy number and deletion fraction were assessed by quantitative qPCR. Additionally, 157 of these fibers were analyzed for sequence variation by deep sequencing.

We found no difference between the PD and control groups in terms of muscle fiber mtDNA copy number or the proportion of molecules containing major arc deletions (Fig. 4a, b and Supplementary Data 8). Likewise, using a linear mixed effects model with study subject as a random effect, we found no association between mtDNA copy number or deletion levels and disease status (Supplementary Tables 13–14). Overall, there was a positive correlation between mtDNA copy number and VDAC1 immunofluorescence (Spearman's $\rho(221) = 0.38$, $P = 8.6 \times 10^{-9}$), indicating that mtDNA copy number reflected mitochondrial content. This was evident in both the PD (Spearman's $\rho(109) = 0.33$, $P = 3.8 \times 10^{-4}$) and control (Spearman's $\rho(110) = 0.39$, $P = 2.5 \times 10^{-5}$) groups. Furthermore, mtDNA copy number showed a positive correlation with the levels of CI (Spearman's $\rho(221) = 0.21$, $P = 0.0020$) and CIV (Spearmans's $\rho(221) = 0.36$, $P = 5.2 \times 10^{-8}$). Single fiber mtDNA deletion levels were generally low (Fig. 4b) and showed no correlation with the levels of CI (Spearman's $\rho(221) = -0.017$, $P = 0.800$) or CIV (Spearman's $\rho(221) = -0.047$, $P = 0.487$).

To determine the presence of point mutations, mtDNA was amplified and sequence variation was assessed in two amplicons covering most of the mtDNA length. This was done using ultra-deep sequencing at a target depth of 100,000x in 157 of the same muscle fibers used for copy number and deletion analyzes (Fig. 4c–f and Supplementary Data 9–10). The two amplicon regions were analyzed separately due to different mean depth of coverage (mean depth amplicon $1:1.66 \times 10^4 \pm 1.01 \times 10^4$; amplicon $2:6.80 \times 10^4 \pm 1.51 \times 10^4$; $P < 10^{-15}$, paired Wilcoxon signed rank test). Single fibers exhibited a median of 14 and 16 heteroplasmic positions at levels above 1% in amplicon 1 and 2, respectively. The heteroplasmic load (i.e., the sum of all heteroplasmic levels across each amplicon) was not associated with disease status, age, sex, or with per-fiber CI levels (Supplementary Table 15).

**Functional complex I deficiency in PD muscle is not associated with mitochondrial DNA variation.** Since the MRC activity measurements had been performed in bulk tissue, we also assessed mtDNA in bulk muscle tissue from 27 individuals, comprising PD with CI activity similar to controls ($n = 8$), PD with low CI activity ($n = 9$), and controls ($n = 10$; Supplementary Fig. 4 and Supplementary Data 5). The three groups were matched for age (Supplementary Table 1). There was no significant difference in mtDNA copy number or deletion fraction between PD individuals with either normal or low CI activity and the control group (Fig. 5a, b and Supplementary Table 16, Supplementary Data 11). Similarly, there was no significant difference between the PD group with normal or low CI activity in terms of mtDNA copy number or deletion fraction (Supplementary Table 16). Compared to the results in single fibers, the analyzes for sequence variation in bulk muscle revealed very few heteroplasmic positions (mean of 0.33 and 1.00 heteroplasmic sites per sample in amplicon 1 and 2, respectively; Fig. 5c, f, Supplementary Data 12, 13). Most samples displayed a single heteroplasmy above 1 % or none at all (26/27 and 19/27 samples showed one or no heteroplasmic positions in amplicon 1 and 2, respectively). There was no significant difference in heteroplasmic load between the three groups (amplicon 1: $P = 0.421$, Kruskal–Wallis $\chi^2 = 1.73$; amplicon 2: $P = 0.111$, Kruskal–Wallis $\chi^2 = 4.40$). Finally, there was no association between heteroplasmic load and CI activity (Supplementary Table 17).

## Discussion

In this work, we sought to explore the prevalence of mitochondrial dysfunction in skeletal muscle of individuals with PD, an area marked by conflicting findings from previous research[15]. Our central hypothesis, based on our recent findings in brain tissue[16], was that MRC dysfunction, specifically in the form of CI deficiency, may occur in a distinct subpopulation of individuals with PD. To increase the power and validity of our analyses compared to earlier research, we employed a much larger cohort ($n = 112$) than previously reported[15], and obtained our samples from a well-characterized cohort of clinically verified PD and neurologically healthy controls.

We show that most individuals with PD exhibit no signs of quantitative or functional MRC alterations in their skeletal muscle. Conversely, MRC deficiency in the form of a functional CI defect occurs in a subset of cases. This subgroup accounted for ~9–16% of our population-based cohort, depending on where the threshold was set. While there was no significant difference in CI levels between PD and controls at the group level, we found a positive association between CI quantity and function. Thus, we cannot exclude that the observed functional defect may be, in part, mediated by a quantitative reduction of the complex. While CII–CIV activities were not significantly different between the PD and control groups, the range of CII and CIV activities in PD extended below that of controls, suggesting a broader involvement of the MRC may occur in PD muscle. Previous studies have also reported variable involvement of other MRC complexes beyond CI, mainly CIII and CIV[48–51]. However, in our cohort, individuals with PD who showed low CI activity generally did not exhibit decreased activity in other complexes (Supplementary Fig. 13). The percentage of individuals with functional CI deficiency in skeletal muscle was lower than the percentage with neuronal CI deficiency in our previous work (~25%)[16]. This discrepancy may arise from several factors. Mitochondrial involvement in skeletal muscle in PD could be mild compared to neuronal tissue, making CI deficiency more difficult to detect. Additionally, given the heterogeneity of PD, larger and more diverse cohorts are likely needed to determine precise and generalizable prevalence estimates for CI deficiency in muscle.

Our findings explain the conflicting nature of previous results. Earlier studies of MRC function in muscle employed small samples sizes, ranging from 3 to 27 individuals with PD[15]. As shown by our data, these were generally underpowered for detecting CI deficiency. In further support of this, several previous studies reporting MRC dysfunction in PD muscle

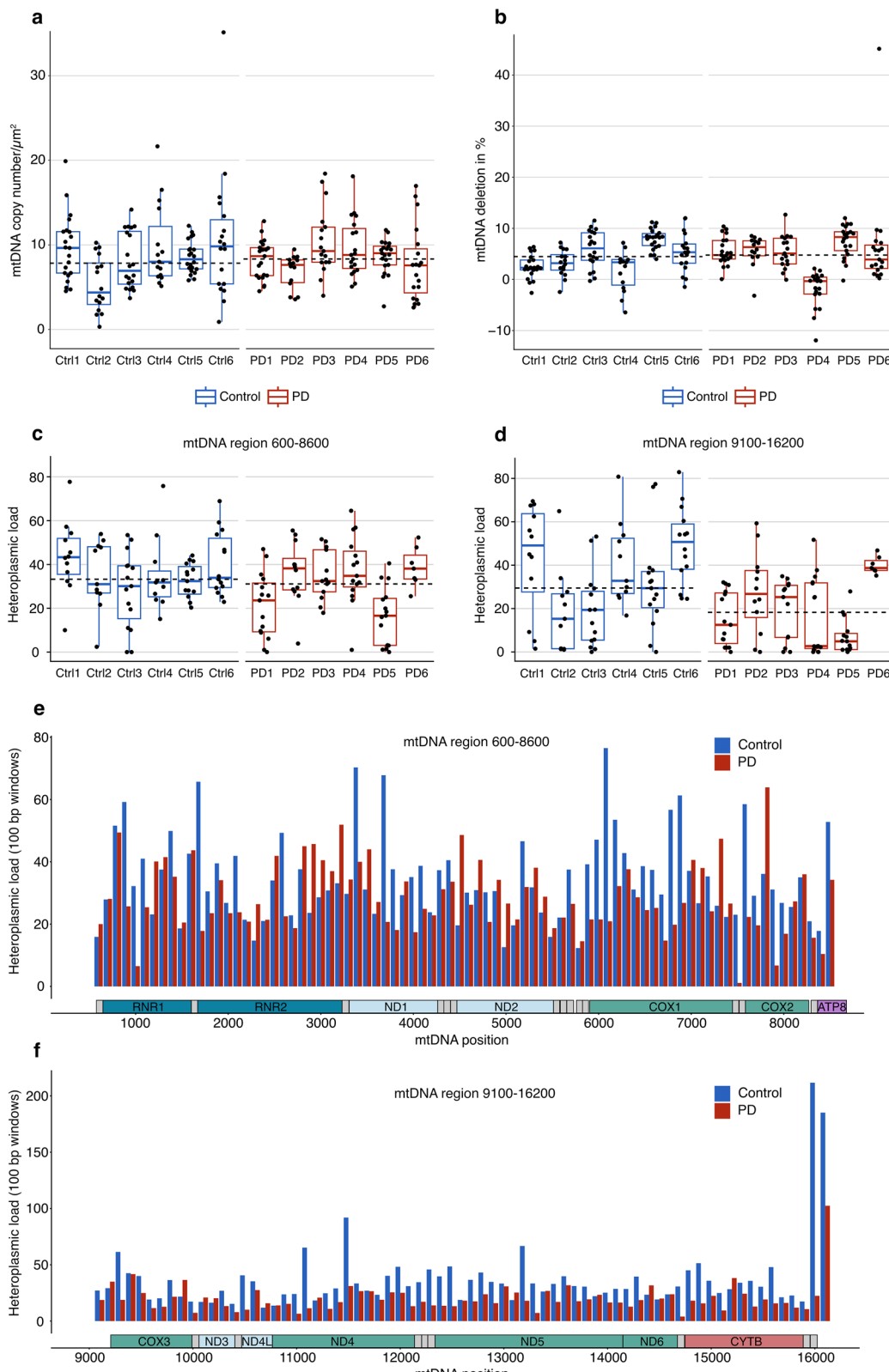

**Fig. 4 | Single muscle fiber mitochondrial DNA profile.** Single muscle fiber mitochondrial DNA (mtDNA) profile from six individuals with PD and six controls. mtDNA copy number and major arc deletion fraction were assessed in a total of 223 muscle fibers, and 157 of the same fibers were analyzed for sequence variation. Red boxplots represent individuals with PD and blue boxplots represent controls. Boxes: median and interquartile range (IQR); whiskers: 1.5x IQR from the lower and upper quartiles. Each dot represents a single muscle fiber. Dashed lines show group-level medians. **a** mtDNA copy number per micro dissected area (μm²) in single muscle fibers (n = 14–22 per individual). **b** Major arc deletion fractions in the same muscle fibers. Deletion fractions were calculated in reference to two samples from blood genomic DNA from healthy controls, which were defined as non-deleted. **c**, **d** Heteroplasmic load in single muscle fibers, defined as the sum of the hetero-plasmy values across the mtDNA region, assessed separately in the regions 600–8600 (amplicon 1) and 9100–16,200 (amplicon 2). Heteroplasmy levels were only considered if above 1% and restricted to single nucleotide variants. **e**, **f** The distribution of heteroplasmy within the mtDNA in the PD (red) and control (blue) groups is shown as the sum of all heteroplasmic levels in 100 bp windows (y-axis) across the mtDNA amplicons (mtDNA coordinates, x-axis).

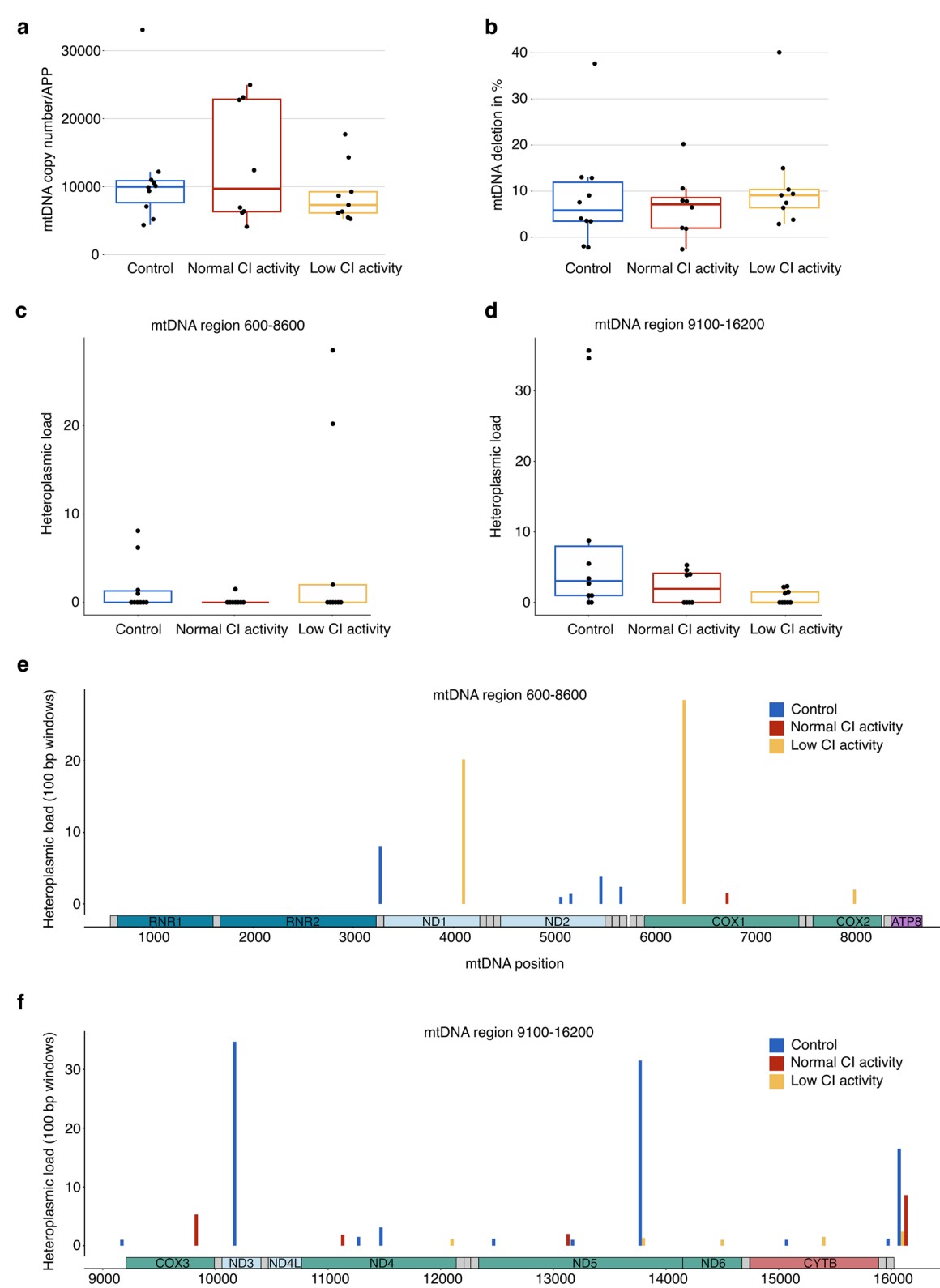

**Fig. 5 | Assessment of association between functional complex I deficiency and mitochondrial DNA changes in PD muscle.** Bulk muscle tissue mitochondrial DNA (mtDNA) analyses from eight PD individuals with complex I (CI) activity similar to controls (red boxplots and bars), nine PD individuals with low CI activity (yellow boxplots and bars), and 10 controls (blue boxplot and bars). The groups were matched for age. Boxes: median and interquartile range (IQR); whiskers: 1.5x IQR from the lower and upper quartiles. Each dot represents one individual. **a** mtDNA copy number in bulk muscle tissue, normalized to the nuclear gene *APP*.

**b** Major arc deletion fractions in bulk muscle tissue. **c, d** Heteroplasmic load in bulk muscle tissue, defined as the sum of the heteroplasmy values across the mtDNA region, assessed separately in the regions 600–8600 (amplicon 1) and 9100–16200 (amplicon 2). Heteroplasmy levels were only considered if above 1% and restricted to single-nucleotide variants. **e, f** The distribution of heteroplasmy within the mtDNA in the three groups is shown as the sum of all heteroplasmic levels in 100 bp windows (*y*-axis) across the mtDNA amplicons (mtDNA coordinates, *x*-axis).

show a considerable overlap between the PD and control groups, with only a subset of cases displaying a clear reduction of CI activity levels, similar to our results[48,49,52]. Technical methodological differences may also contribute to variability among studies[15].

The notion that skeletal muscle mitochondrial function is impaired only in a small subset of individuals with PD is further supported by previous in vivo [31]phosphorus magnetic resonance spectroscopy ([31]P-MRS) studies. A study in forearm muscle reported a high (>2 SDs above the mean of controls) inorganic phosphate/phosphocreatine (Pi/PCR) ratio, indicative of impaired mitochondrial bioenergetic status, in a subset of 9 out of 28 individuals with PD (32%)[53]. Additionally, evidence of mitochondrial dysfunction in the form of lower maximum mitochondrial ATP production (ATPmax) has been found by [31]P-MRS in the tibialis anterior muscle of individuals with PD[54]. Notably, ATPmax displayed substantial overlap with the control group, with only 2 out of 29 individuals with PD (7 %) displaying levels <2 standard deviations of the mean of the controls.

Studies in other extra-neural tissues have also shown highly variable results, with some reporting MRC deficiencies and some not. A thorough assessment of the available data reveals a similar picture to our results in skeletal muscle, i.e., a substantial overlap between the PD and control groups, with a minority of PD cases falling below the range of controls. Studies in platelets that report deficiency of CI activity show a variable decrease of ~16–51% when comparing PD and controls at the group level, but with a considerable overlap between the PD and control groups, suggestive of a subgroup with deficiency[55–59]. In contrast to our analyses in skeletal muscle, some of these studies report functional impairment in the other MRC complexes as well[56,58,59]. Studies in skin fibroblasts have shown similar heterogeneity, with impaired mitochondrial function present only in a subset of cases[60–62]. One of these studies reported indirect evidence of impaired mitochondrial respiration by means of decreased mitochondrial membrane potential in a subset of 5% of their PD cohort, using two standard deviations of the control group as a reference level of normal mitochondrial membrane potential[60]. Studies in PD lymphocytes that report mitochondrial dysfunction have been contradictory, with one study reporting CI deficiency in 3/16 individuals with PD (19%)[63], while another showed decreased CI activity in all the analyzed (20/20) individuals with PD[64]. These studies reported variable deficiency of CIV as well.

Whether the observed reduction of CI activity in the muscle of individuals with PD relates to the primary pathophysiology of the disease or is a secondary phenomenon induced by factors such as altered mobility or drug treatment cannot be ascertained by our study. However, the fact that we did not observe a significant association between CI activity and disease duration suggests that it is not secondary to immobility or dopaminergic treatment. The latter is supported by a study in platelets reporting no effect of the initiation of carbidopa/levodopa and selegiline treatment on CI activity[65], and in rat skeletal muscle, showing that MRC activity was not altered by levodopa treatment[66]. In line with this, CI deficiency has also been reported in platelets and lymphocytes of drug naïve individuals with PD[59,63]. While we did not detect a correlation between muscle CI deficiency and disease progression or clinical phenotype, larger cohorts with longitudinal follow-up are necessary to evaluate whether such correlations exist. These questions will be explored in the ongoing STRAT-PARK study[19]. Even in the absence of clear clinical correlates, identifying molecular subtypes can have therapeutic benefits by enabling individuals to participate in mitochondria-targeted therapy trials. This approach reflects a broader shift toward bottom-up molecular stratification as a more precise framework for personalized therapies in PD[6].

The mechanisms mediating functional CI deficiency in PD muscle remain unknown. Although we found a positive association between CI quantity and function, we did not detect a significant reduction of CI quantity in muscle tissue of individuals with PD. This is in contrast to the PD brain, which demonstrates both a quantitative reduction and functional deficiency in CI[67]. Moreover, the dysfunction is not attributable to qualitative or quantitative alterations in mtDNA, as our analysis showed no correlation with mtDNA copy number, deletion levels, or point mutations.

One contributing factor may be genetic variation in one or more of the 38 nuclear-encoded CI subunits and/or factors required for CI assembly. No individual variants[68], or polygenic enrichment of rare coding variants[69,70] in these genes have been associated with PD. However, an association of PD with a high polygenic score of common variants in genes related to oxidative phosphorylation (OXPHOS) was recently reported[71]. This polygenic score was found to be associated with altered respiratory function in fibroblasts and induced pluripotent stem cell-derived neuronal progenitors from patients, but no specific CI defect was shown. Furthermore, individuals with high OXPHOS polygenic score had an earlier age of onset, a feature not observed in our CI-deficient subgroup. Our findings could be attributable to exposure to environmental inhibitors of CI. Adequately characterizing the exposure to CI inhibitors is challenging, as many diverse natural and synthetic compounds are known inhibitors[72]. While our results did not indicate a connection with agricultural work or a documented history of pesticide exposure, the possibility of dietary pesticide exposure remains unaccounted for. Given the established link between pesticide exposure and PD[73], further investigation into this area is warranted[74].

Mitochondrial function, including CI function, has been shown to decline with age in skeletal muscle[75–77]. In our cohort, there was a mild and significant association between increasing age and lower CI and CIV levels, but not activities. This may be due to the higher age range of our cohort (45–84 years), compared to those employed in aging studies, which extend into the early 20s[75–77]. Whether the observed CI deficiency in PD muscle reflects an impaired capacity to compensate for age-related mitochondrial decline remains unknown. Another important consideration is the potential role of impaired proteostasis and α-synuclein pathology. Studies in cell and animal models indicate that misfolded α-synuclein can impair mitochondrial function, including CI activity[78–80], and mitochondrial dysfunction has been reported in skeletal muscle of mice overexpressing α-synuclein[81]. However, whether α-synuclein pathology occurs in PD muscle and, if so, is linked to mitochondrial dysfunction, is currently unexplored. This is a pertinent area for future research.

Our study has certain limitations that must be taken into consideration. The needle biopsy approach used to obtain the muscle samples has the benefit of reduced invasiveness, compared to an open muscle biopsy, making it feasible to perform on the large number of subjects employed in our study. The procedure was well-tolerated with a brief convalescence (avoiding weight-bearing on the biopsied leg for 1 h and refraining from strenuous exercise for 48 h), and local complications were rare and limited to minor hematomas. However, due to the relatively low amounts of biopsy material obtained with this method, we were not able to perform both the immunohistochemistry and enzymatic activity assays in all study subjects (Supplementary Data 5). While the PD and control groups were well matched for age, they were unbalanced in terms of sex, with a male-to-female ratio of ~1.9 in the PD group and ~0.4 in the control group. This is mainly because PD has a higher prevalence in males[82], and most control individuals were recruited among the spouses of the individuals with PD. Since our immunohistochemistry assay targeted individual subunits of CI and CIV, we cannot exclude defects of complex/supercomplex assembly and/or altered subunit composition as the cause of deficient function. Investigating this would require analyzing the assembly status of the MRC complexes using methods such as blue native gel electrophoresis and complexome profiling[83,84]. In this study, it was not possible to perform these additional analyzes due to the limited quantity of muscle available to us. Finally, the individuals with PD included in this project have not been genetically characterized. The prevalence of monogenic PD in Norway is very low, estimated at ~0.5% and virtually accounted for by *LRRK2* mutations, based on a population-representative cohort from the region of origin of the majority of our samples (Western Norway)[85]. Thus, it is highly unlikely that any substantial number of monogenic PD cases were included in our cohort. *GBA* variation, which is more common[86], may be of relevance, and this should be explored in future studies. Moreover, since the original neuronal CI-deficient sub-population of PD was found in a genetically characterized sample of

idiopathic PD, it is generally unlikely that this subgroup is driven by monogenic contributions.

In summary, regardless of its etiology and pathogenic contribution, our findings confirm the hypothesis that CI deficiency in skeletal muscle is not a pervasive feature of PD, but one that occurs only in a subset of cases. Whether these are the same as (or overlap with) the ~25% of the PD cases with widespread neuronal CI deficiency in the post-mortem brain[16], remains to be determined. Interestingly, individuals with PD exhibiting low CI activity in muscle were predominantly females, a trend also exhibited by the subgroup with neuronal CI deficiency[16]. However, unlike the subgroup with neuronal CI deficiency, individuals with low muscle CI activity did not exhibit a predilection for a non-tremor dominant motor phenotype. Determining whether CI deficiency in PD muscle corresponds with neuronal CI deficiency in the brain will require examination of muscle and brain tissue from the same individuals. To the best of our knowledge, such a material is not currently available. However, the vast majority of the individuals with PD and controls in the STRAT-PARK cohort[19], where most of our muscle samples originate from, have consented to post-mortem brain donation. Thus, we hope to answer this pertinent question in the future. In the meantime, the applicability of muscle CI deficiency as a clinical stratification biomarker is limited by the invasiveness of the muscle biopsy, even with our percutaneous technique[19]. Examination of the respiratory chain integrity in more easily accessible samples, such as blood platelets or PBMCs, is warranted to establish whether classification of PD according to mitochondrial pathology can be achieved with more efficient and less invasive testing.

## Data availability

The immunohistochemistry, enzymatic activity, and mtDNA qPCR data generated in this study, as well as the source data for Figs. 1–5, are provided in the Supplementary Data files available at figshare: https://doi.org/10.6084/m9.figshare.c.7581485.v3. The sequencing data of the bulk tissue samples and the single muscle fiber samples are available in the Federated European Genome-phenome Archive (FEGA) Norway (accession number: EGAD50000000946). Other data are available from the corresponding author on reasonable request.

## Code availability

The R code required to reproduce the results of the statistical analyses is available in GitLab[87]: https://git.app.uib.no/simon.kverneng/complex-i-deficiency-in-skeletal-muscle-of-a-subgroup-of-parkinsons-disease.

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

## Acknowledgements

We are grateful to the study participants and their families for their essential contribution to the study. We thank the STRAT-PARK study teams at Haukeland University Hospital and St. Olav's University Hospital, including research nurses Erika Sheard, Mona Søgnen, Solveig Af Geijerstam, Therese Vetås, and Anne Grete Wahlvåg, as well as laboratory personnel Martina Castelli, Yana Mikhaleva, Gry Hilde Nilsen, Omnia Shadad, Dagny Ann Sandnes, and Ida Johansson for the outstanding technical support. We are also grateful to the STRAT-COG study team, including Kristina Skeie and Lone Birkeland Johnsen, and the NADPARK study team, including research nurse Marit Renså. We thank Nelson Osuagwu for technical assistance with fluorescence image acquisition. The fluorescence imaging was performed at the Molecular Imaging Center, Department of Biomedicine, University of Bergen. This work is supported by grants from The Michael J Fox Foundation (MJFF-022567; C.T.), The Research Council of Norway (288164; C.T.), The KG Jebsen Foundation (SKGJ-MED-023; C.T.), and the Western Norway Regional Health Authority (F-12598-D12092; S.U.K.).

## Author contributions

S.U.K.: participated in study conception and design, led the study, recruited and assessed study participants, collected, analyzed, and interpreted data, and drafted the manuscript. K.E.S.: participated in study conception and design, recruited and assessed study participants, and collected and interpreted data. H.B., K.L., B.B., G.O.S., R.E.S., K.H.: recruited and assessed study participants, collected and interpreted data. S.M., M.B., E.F.V.: collected and interpreted data and participated in the drafting of the manuscript. I.F.: participated in data collection and interpretation. L.T.: Participated in study design and advised on statistical approaches. Y.N.T.C.: participated in study conception and funding acquisition. C.D.: participated in study conception, design, data interpretation, and data collection. G.S.N.: generated, analyzed, and interpreted data and drafted the manuscript. C.T.: conceived, designed, and directed the study, contributed to data analyzes and interpretation, drafted the manuscript, and acquired funding for the study. All authors have read and approved the manuscript.

## Funding

## Competing interests

The authors declare no competing interests.
