## [Transparent Peer Review file · Communications Medicine]

Mitochondrial complex I deficiency occurs in skeletal muscle of a subgroup of individuals with Parkinson's disease

Corresponding Author: Professor Charalampos Tzoulis

This manuscript has been previously reviewed at another journal. This document only contains information relating to versions considered at Communications Medicine.

Version 0:
Reviewer comments:

Reviewer #1

(Remarks to the Author)

The authors fulfilled all the reviewer requests and the revised version of the manuscript has been substantially improved. The manuscript is now ready for publication in its current format

Reviewer #2

(Remarks to the Author)

The authors have addressed the majority of the previous comments either by text or by figure changes. There are only 2 minor comments which should be addressed:

Minor comments:

1. All the comments made in Response 1 (to Reviewer #5) (See 3 Sections: Percentage of PD patients with Complex I (CI) deficiency in muscle; Correspondence between neuronal and muscle CI activity; Relevance of decreased muscle CI activity to patient symptoms and disease progression) should be added to the manuscript's discussion.
2. Supp Fig 3: There should be a white line drawn (or a space) between left images and right images (since currently unclear where the images start/end)
- 3.

Reviewer #3

(Remarks to the Author)

Reviewer #4

(Remarks to the Author)

My prior comments still stand: In this manuscript, Kverneng et al., have tested the hypothesis that Parkinson's disease (PD) patients are characterized by mitochondrial complex I (CI) deficiency that can be revealed by analyses of peripheral tissues such as skeletal muscles through minimally invasive procedures. They report that a subset of PD patients exhibit decreased CI activity in the skeletal muscle. This defect was isolated to CI as deficiencies in other complexes of the respiratory chain were not observed. In addition, the authors ruled out the possibility that the decreased CI activity is due to mtDNA mutations in these patients. The methods are described in detail, including descriptions of detailed statistical analyses. Notably, the discussion is appropriately conservative and careful in its conclusions while also fairly emphasizing the paper's strengths.

This work contributes significantly to the field of PD diagnosis for two reasons. Firstly, this study employs the largest cohort of PD patients that have been analyzed for skeletal muscle tissue (n=83) and respiratory enzymatic activities (n=57), besides the 27 PD patients that were analyzed by Blin et al., 1994. Considering the inconsistencies in skeletal muscle CI deficiency observed in the past, this study highlights the importance of analyzing larger cohorts to glean meaningful information for the PD community. Secondly, this study also underscores the importance of the method of detection – as clearly there was a measurable decrease in CI-specific activity with no significant difference in CI abundance. Lastly, as interests focus more in idiopathic disease, patient stratification for clinical trials, and clinical sub-typing, these biochemical parameters for sub-typing may become very important.

The authors have adequately addressed concerns.

UNIVERSITY OF BERGEN

Department of Clinical Medicine

Department of Neurology

Bergen, 28.02.25

We are grateful to the Reviewers for their comments and have addressed these in our point-by-point response below, and in the manuscript. Additionally, we have addressed the editorial comments and requests in the revised manuscript. Finally, a few minor errors have been corrected which have absolutely no impact on the results. All changes in the manuscript and supplementary information have been marked using the “Track Changes” function of Microsoft Word.

Yours sincerely,

Charalampos Tzoulis, MD, PhD
Professor of Neurology and Neurogenetics
Head, Neuromics Research Group
Director, K.G Jebsen Center for Translational Research in Parkinson’s disease
Director, Neuro-SysMed National Center of Excellence for Neurological Diseases
Department of Neurology, Haukeland University Hospital
Department of Clinical Medicine, University of Bergen
<http://www.neuromics.org/>
charalampos.tzoulis@nevro.uib.no
Tel: +4755975045
Fax: +4755975164

UNIVERSITY OF BERGEN

Department of Clinical Medicine

Department of Neurology

Bergen, 28.02.25

Reviewer #1

Comment 1:

The authors fulfilled all the reviewer requests and the revised version of the manuscript has been substantially improved. The manuscript is now ready for publication in its current format

Response-1:

We thank the reviewer for the positive evaluation of our work.

Reviewer #2

The authors have addressed the majority of the previous comments either by text or by figure changes.

There are only 2 minor comments which should be addressed:

Comment 1:

1. All the comments made in Response 1 (to Reviewer #5) (See 3 Sections: Percentage of PD patients with Complex I (CI) deficiency in muscle; Correspondence between neuronal and muscle CI activity; Relevance of decreased muscle CI activity to patient symptoms and disease progression) should be added to the manuscript's discussion.

Response 1:

We thank the reviewer for these comments and have added this information in the discussion.

Comment 2:

2. Supp Fig 3: There should be a white line drawn (or a space) between left images and right images (since currently unclear where the images start/end)

Response 2:

We have amended the figure as requested.

Reviewer #4

My prior comments still stand: In this manuscript, Kverneng et al., have tested the hypothesis that Parkinson's disease (PD) patients are characterized by mitochondrial complex I (CI) deficiency that can be revealed by analyses of peripheral tissues such as skeletal muscles through minimally invasive procedures. They report that a subset of PD patients exhibit decreased CI activity in the skeletal muscle. This defect was isolated to CI as deficiencies in other complexes of the respiratory chain were not observed. In addition, the authors ruled out the possibility that the decreased CI

UNIVERSITY OF BERGEN

Department of Clinical Medicine

Department of Neurology

Bergen, 28.02.25

activity is due to mtDNA mutations in these patients. The methods are described in detail, including descriptions of detailed statistical analyses. Notably, the discussion is appropriately conservative and careful in its conclusions while also fairly emphasizing the paper's strengths.

This work contributes significantly to the field of PD diagnosis for two reasons. Firstly, this study employs the largest cohort of PD patients that have been analyzed for skeletal muscle tissue (n=83) and respiratory enzymatic activities (n=57), besides the 27 PD patients that were analyzed by Blin et al., 1994. Considering the inconsistencies in skeletal muscle CI deficiency observed in the past, this study highlights the importance of analyzing larger cohorts to glean meaningful information for the PD community. Secondly, this study also underscores the importance of the method of detection – as clearly there was a measurable decrease in CI-specific activity with no significant difference in CI abundance. Lastly, as interests focus more in idiopathic disease, patient stratification for clinical trials, and clinical sub-typing, these biochemical parameters for sub-typing may become very important.

The authors have adequately addressed concerns.

Response:

We thank the reviewer for the insightful comments and for the positive evaluation of our work.

Correction of minor errors:

MDS-UPDRS III scores

An error in the summation of MDS-UPDRS III score was discovered in a subset of individuals. This has been corrected in Table 1, as well as in Supplementary Table 8 and 9, and the relevant Supplementary Data files. Results of the analyses remain unchanged.

Figure 2e-f

The previous version of this figure erroneously did not show batch-adjusted data for VDAC1. The figure has been revised to show batch-adjusted data.

Figure 4c-d and Supplementary Table 15.

Single fibers with heteroplasmic load of “0” were erroneously not included in the previous version of Figure 4c-d and in the linear mixed effects model of heteroplasmic load in single muscle fibers (Supplementary Table 15). This has been corrected. The result of the analysis remains unchanged.